# AutoHall: Automated Hallucination Dataset Generation for Large Language Models

## Abstract

While Large language models (LLMs) have garnered widespread applications across various domains due to their powerful language understanding and generation capabilities, the detection of non-factual or hallucinatory content generated by LLMs remains scarce. Currently, one significant challenge in hallucination detection is the laborious task of time-consuming and expensive manual annotation of the hallucinatory generation. To address this issue, this paper first introduces a method for automatically constructing model-specific hallucination datasets based on existing fact-checking datasets called **AutoHall**. Furthermore, we propose a zero-resource and black-box hallucination detection method based on self-contradiction. We conduct experiments towards prevalent open-/closed-source LLMs, achieving superior hallucination detection performance compared to extant baselines. Moreover, our experiments reveal variations in hallucination proportions and types among different models.

## 1 Introduction

Large language models (LLMs) such as ChatGPT [1], GPT-4 (OpenAI, 2023), Claude (Bai et al., 2022) and Llama-2 (Touvron et al., 2023) have achieved widespread popularity and adoption across diverse industries and domains (Sohail et al., 2023; Sallam, 2023; Sallam et al., 2023). Despite their powerful capabilities, the issue of "hallucination" poses a concern that LLMs have the tendency to generate inaccurate/fabricated information in generation tasks (Zhang et al., 2023b; Ji et al., 2023). As shown in Fig. 1, ChatGPT suffers from hallucination when giving a description of the novel "The Leopard" by Norwegian author Jo Nesbø. It can be observed that ChatGPT makes up some plots of the novel and contains incorrect texts in the response, because the novel never mentions the presence of a "red diamond" at the crime scene and the "The Snowman" case has also been solved before. Since the current artificial intelligence relies more on LLMs, hallucinatory information indeed disturbs the enterprise security and the user trust (Zhang et al., 2023a; Gupta et al., 2023). Therefore, detecting hallucinations generated by the LLMs is of significant importance.

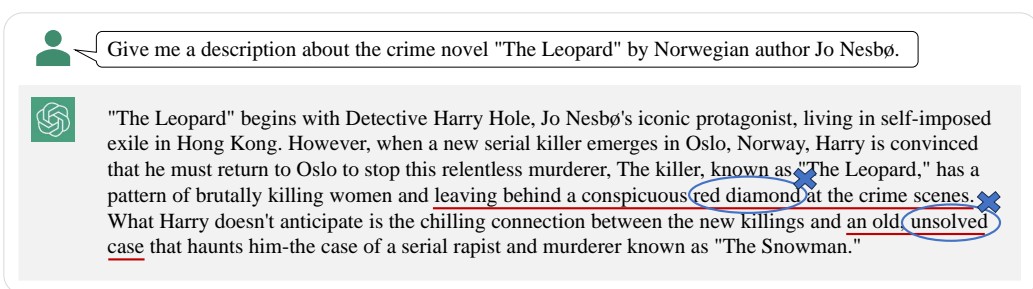

Figure 1: A hallucination example. The red underline indicates the hallucinatory content.

Current research efforts on hallucination detection leverage external knowledge sources (Chern et al., 2023; Gou et al., 2023) or just adopt a zero-resource approach, which focuses on resources

---

[1]https://chat.openai.com/

inherent to the model itself (Azaria & Mitchell, 2023; Agrawal et al., 2023; Varshney et al., 2023; Manakul et al., 2023b; Mündler et al., 2023). Typically, most of these methods begin with a crowd-sourced annotation, where researchers use QA datasets to have the model generate responses and then manually annotate whether the answers contain hallucinations.

However, these sort of model-specific "hallucination detection" datasets all have their own limitations. For one thing, each model requires a full annotation of the dataset. For another, such a dataset is also time-sensitive as upgrades may mitigate hallucination issues in LLMs and the old dataset is no longer applicable to the new model.

Considering the above issues, this paper explores one automated generation of hallucination detection datasets. Inspired by Agrawal et al. (2023) emphasizing the hallucinatory reference problem in LLMs, we find the possibility of automatically creating hallucination detection datasets through public fact-checking datasets. Specifically, since the existing fact-checking datasets usually consist of manually annotated claims accompanied by the ground truth labels (i.e., factual/unfactual), we can determine whether hallucination has occurred by generating references to the claims and exploring whether the references can infer the correct labels for the claims.

In addition, we further propose a three-step zero-resource black-box hallucination detection method based on our dataset inspired by the idea of self-contradictory (Wang et al., 2022; Mündler et al., 2023; Manakul et al., 2023b). Given an LLM accurately understands one claim, its randomly sampled references are less likely to contain contradictions. Therefore, it is possible to determine whether the model has generated hallucinations based on knowledge conflicts among these references. In summary, the contributions of our paper are:

- We propose an approach for fast and automatically constructing model-specific hallucination datasets based on existing fact-checking datasets called **AutoHall**, eliminating the need for manual annotation.

- Based on our dataset, we introduce a novel black-box hallucination detection method without external resources. Then, we evaluate its effectiveness on ChatGPT and Llama-2 models, demonstrating its superior improvements over existing detection techniques.

- From the analysis of our experimental results, we estimate the prevalence of hallucination in LLMs at a rate of 20% to 30% and gain insight into what types or topics of LLM responses that tend to be hallucinatory.

## 2 RELATED WORKS

### 2.1 HALLUCINATION OF LARGE LANGUAGE MODELS

Although large language models have demonstrated remarkable capabilities (Liu et al., 2023; Srivastava et al., 2022), they still struggle with several issues, where hallucination is a significant problem. Hallucination arises when the content generated by LLMs is fabricated or contradicts factual knowledge. The consequent effects may be harmful to the reliability of LLM applications (Zhang et al., 2023b; Pan et al., 2023). So far, the causes of hallucination in LLMs have been investigated across different tasks, such as question answering (Zheng et al., 2023), abstractive summarization (Cao et al., 2021) and dialogue systems (Das et al., 2023). The key factors include but are not limited to training corpora quality (McKenna et al., 2023; Dziri et al., 2022), problematic alignment process (Radhakrishnan et al., 2023; Zhang et al., 2023b) and randomness in generation strategy (Lee et al., 2022; Dziri et al., 2021).

### 2.2 LLM HALLUCINATION DETECTION

To detect the hallucination issue, there are many endeavors to seek solutions. On the one hand, prior works focus on resorting to external knowledge to detect hallucinations. For instance, Gou et al. (2023) propose a framework called CRITIC to validate the output generated by the model with tool-interaction and Chern et al. (2023) invoke interfaces of search engines to recognize hallucination. On the other hand, current research pays more attention to realizing one zero-resource hallucination detection method. Typically, Xue et al. (2023) utilize the Chain of Thoughts (CoT) to check the

hallucinatory responses; Manakul et al. (2023b) introduce a simple sampling-based approach that can be used to detect hallucination with token probabilities.

Besides, some hallucination benchmarks (Li et al., 2023; Umapathi et al., 2023; Dale et al., 2023) are constructed to support detection tasks in numerous scenarios. For example, Umapathi et al. (2023) propose a hallucination benchmark within the medical domain as a tool for hallucination evaluation and mitigation; Dale et al. (2023) present another dataset with human-annotated hallucinations in machine translation to promote the research on translation pathology detection and analysis.

Nevertheless, there are limitations as they are subject to manually annotated hallucination datasets, which are expensive and time-consuming. Meanwhile, the datasets are model-specific, requiring separate annotations for different models, whose applicability will also be affected by model upgrades. Furthermore, there is also room for improvement in the performance of current hallucination detection methods.

## 3 METHODOLOGY

In this section, we first formulate the definition of LLM hallucination discussed in our work. Then, we introduce our automatic dataset creation pipeline which focuses on prompting LLMs to produce "hallucinatory references". Finally, based on our generated datasets, we further present one zero-resource, black-box approach to recognize hallucination.

### 3.1 LLM HALLUCINATION

LLM hallucination can be categorized into different types (Galitsky, 2023), such as hallucination based on dialogue history, hallucination in generative question answering and general data generation hallucination. They can all be attributed to the generation of inaccurate or fabricated information.

Generally, for any input sentence $X = [x_1, x_2, \ldots, x_n]$ with a specific prompt $P = [p_1, p_2, \ldots, p_o]$, the large language model $\mathcal{M}$ will generate an answer $Y = [y_1, y_2, \ldots, y_m]$, denoted as:

$$\mathcal{M}(P, X) = Y. \tag{1}$$

Given factual knowledge $F = [f_1, f_2, .., f_t]$, the problem of hallucination $H$ occurs when there is a factual contradiction between the output span $Y_{[i:j]} = [y_i, y_{i+1}, \ldots, y_j]$ and the knowledge span $F_{[u:v]} = [f_u, f_{u+1}, \ldots, f_v]$, which can be summarized into the function below:

$$Y \in H \Leftrightarrow \exists Y_{[i:j]} \exists F_{[u:v]}((Y_{[i:j]} \wedge F_{[u:v]} = \text{False})). \tag{2}$$

### 3.2 AUTOHALL: AUTOMATIC GENERATION OF HALLUCINATION DATASETS

Current research on hallucination detection mostly relies on manually annotated datasets. Namely, whether $Y$ is hallucinatory requires slow and costly manual tagging due to the absence of a comparison standard for the factuality. However, the fact-checking datasets provide us with data typically comprising real-world claims, corresponding ground truth labels, and evidence sentences as shown in Fig. 2. We can prompt a model to generate relevant references for claims and then use the ground truth labels as criteria to assess the hallucinatory nature of the generated references. Specifically, as shown in Fig. 2, **AutoHall** generates hallucination datasets following the below three steps:

**Step 1: References Generation.** For an LLM, we prompt it to generate corresponding references to the claims in the existing datasets by the prompt illustrated in Fig. 2 Step 1. Note that to simplify the generation, we only focus on factual (supported/true) and faked (unsupported/false) claims. Besides, we discard references that fail to contain concrete content, like a long response beginning with "I can not provide a specific reference for the claim you mentioned...". The remaining valid references are either reliable ($\overline{H}$) or hallucinatory ($H$).

**Step 2: Claim Classification.** Separately for each reference, in order to label whether a claim belongs to $\overline{H}$ or $H$, we prompt LLM to perform claim classification. The input sequence is of format as shown in Fig. 2 Step 2, where the two placeholders $\langle claim \rangle$ and $\langle reference \rangle$ should be replaced with the claim $X$ and the generated reference $Y$ in Step 1. Then the output is of format

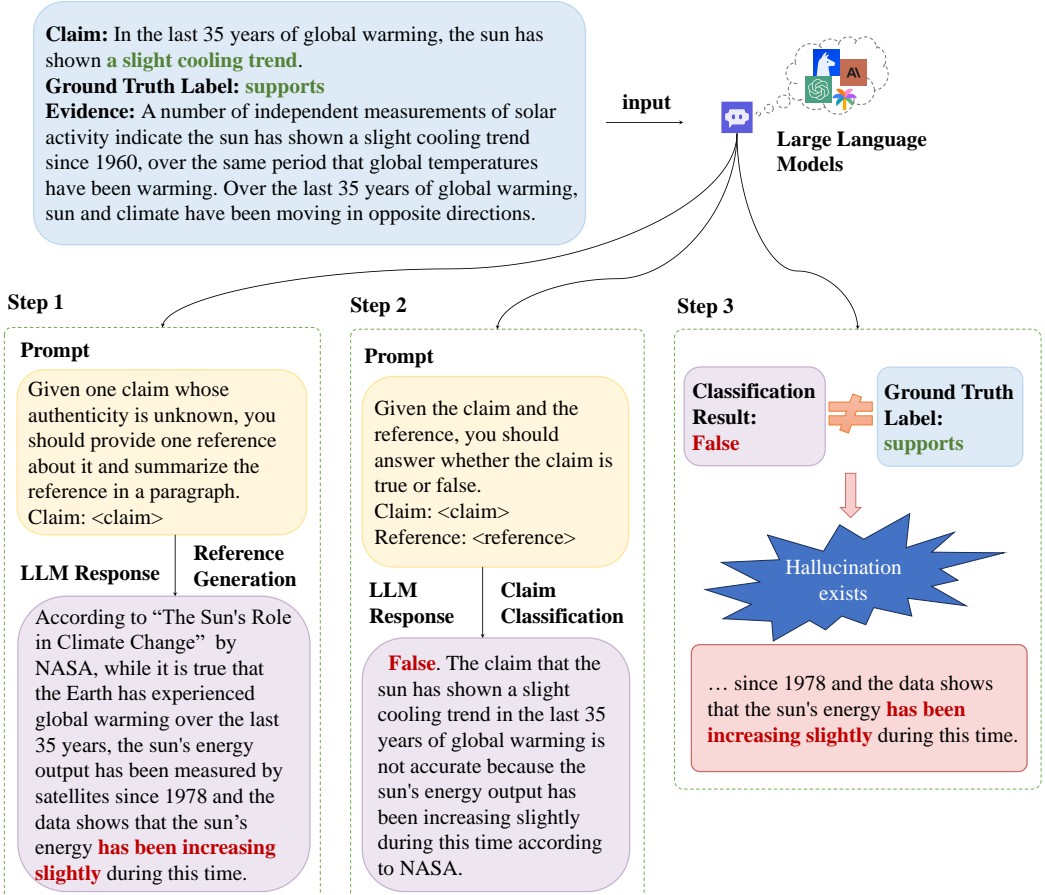

Figure 2: Our proposed approach to collect LLM hallucination automatically. **Green**: the grounded information. **Red**: the incorrect information. The complete prompts are shown in Appendix A and some analysis on prompt sensitivity is included in Appendix B.

"Category: $\langle category \rangle$ Reason (Optional): $\langle reason \rangle$" where the category is limited to true ($T$) or false ($F$). To elaborate, $T$ indicates the generated reference $Y$ supports the claim $X$ is factual and $F$ represents that $Y$ demonstrates claim $X$ is faked. We expect correct classification to each claim, while wrong classification may be taken as a sign of the existence of hallucination in the generated reference that it erroneously supports the claim's factuality. The binary classification results of LLMs are reliable as LLMs exhibit strong capabilities in natural language inference (Wu et al., 2023) and human evaluation gives a guarantee as shown in Appendix C.

**Step 3: Hallucination Collection.** Last, we can directly adopt a simple comparison to collect the hallucination dataset. If the classification result is not equal to the ground truth label, we label the reference as hallucination. Meanwhile, to maintain a balanced proportion between hallucinatory and factual references, we sample the same number of factual references built upon hallucinatory ones to form a completed dataset.

### 3.3 HALLUCINATION DETECTION APPROACH

The rationale for our detection approach is that if the LLM knows one claim well, even when we query it to provide multiple references, self-contradictions among them should be absent otherwise hallucination information must exist in one reference. Compared to SelfCheckGPT (Manakul et al., 2023b), our method uses the LLM for hallucination detection end-to-end rather than relying on output token probabilities to calculate hallucination score with BERTScore or n-gram.

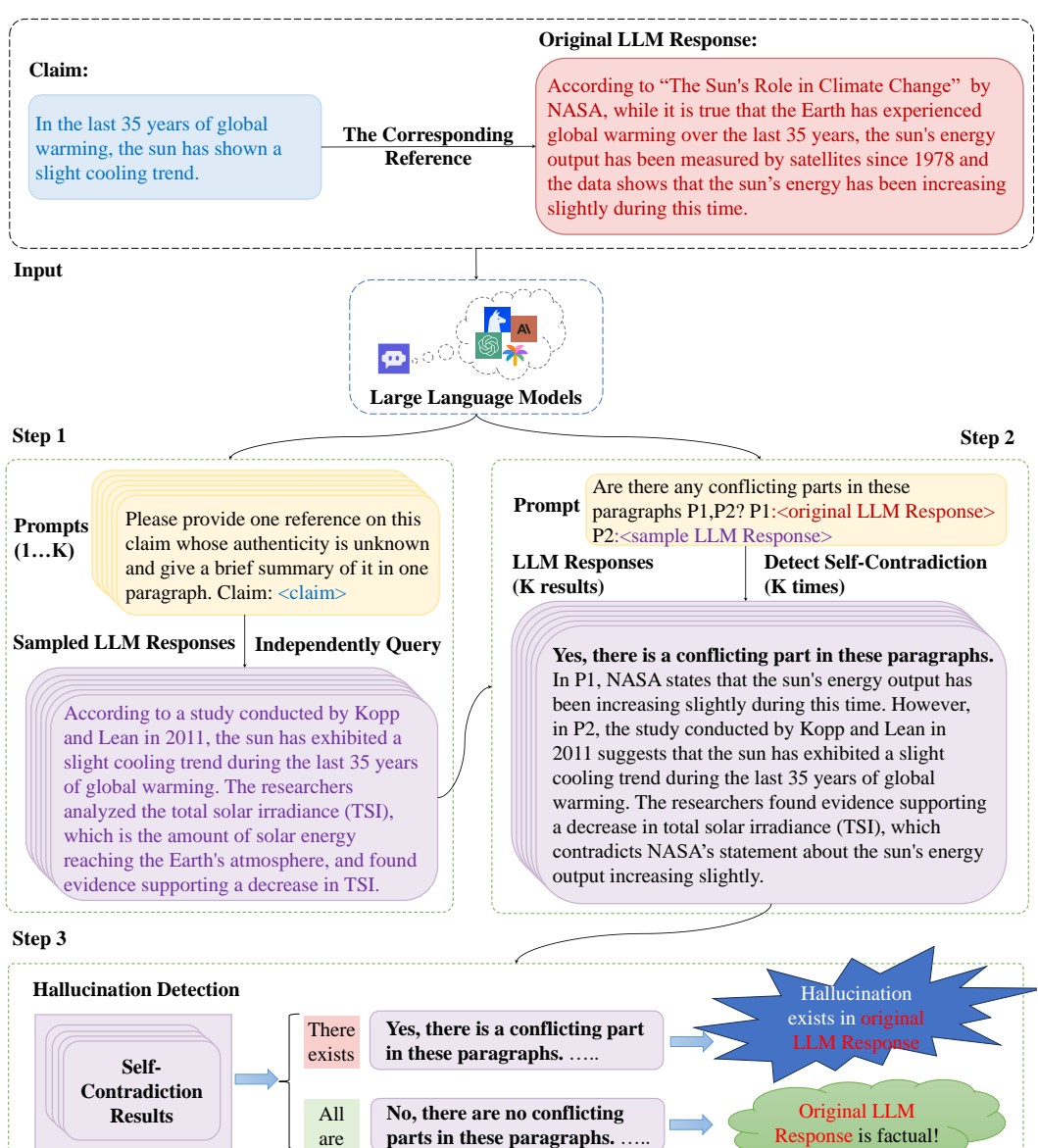

Figure 3: Our proposed approach to detect LLM hallucination. Blue: the claim from fact-checking dataset. Red : the response need to be detected whether exists hallucination. Purple: the sampled references to trigger self-contradictions. The complete Step 2 prompts are shown in Appendix A.

As shown in Fig. 3, to trigger self-contradictions, we first appropriately prompt an LLM to answer a second reference $Y'_k$ and repeat this process $K$ ($K = 13$ in experiments) times. It is worth noting that each query is running independently with an equivalent prompt. Then, we concatenate each generated reference $Y'_k (k = 1, ..., K)$ with the original reference $Y$ to form one input pair. Unlike SelfCheckGPT measures the consistency between $Y$ and all $K$ sampled references, we invoke the LLM to detect if $Y$ and $Y'_k$ are contradictory. Such self-contradiction detection in $\langle Y, Y'_k \rangle$ pair can focus more on the hallucination detection in $Y$ and avoid the problem that SelfCheckGPT incorrectly identifies the conflicts in the $K$ sentences generated subsequently as the hallucination in $Y$.

Formally, we can check if there exists at least one $Y'_k$ conflicting with $Y$, as shown in Eq. (3). If conflicts are indicated, it suggests the model does not understand the claim well, and $Y$ may be hallucinatory. Conversely, if no conflicts are found in $K$ pairs, it indicates that the factual reference.

$$Y \in H \Leftrightarrow \exists Y'_{k,[u,v]} \exists Y_{[i,j]}((Y_{[i,j]} \wedge Y'_{k,[u,v]} = \text{False})) \tag{3}$$

## 4 EXPERIMENTS

### 4.1 EXPERIMENTAL SETTINGS

#### 4.1.1 MODELS

We conduct experiments towards the state-of-the-art open-/closed-source LLMs. For the closed-source model, we select ChatGPT, which is widely recognized as one of the leading closed-source LLMs, with the assistance of paid gpt-3.5-turbo API. We also choose Llama-2-chat (the instruction-tuned version) for the open-source LLM experiments, as it is one of the most prominent open-source models available. Based on our computing resources, we primarily run its 7B&13B parameters versions on a server with dual Nvidia A100 80GB GPUs.

#### 4.1.2 DATASETS AND METRICS

For hallucination collection, we employ three fact-checking datasets: Climate-fever (Diggelmann et al., 2020), Pubhealth (Kotonya & Toni, 2020) and WICE (Kamoi et al., 2023). All of them provide real-world claims, ground truth labels and evidence retrieved from websites as shown in Table 1. The topics of claims range from different domains, such as technology, culture, health and so on, which facilitates our analysis of what types or topics of content LLMs tend to be hallucinatory.

To investigate the hallucination properties of large language models at different temperatures, we set their temperature values as 0.1, 0.5 and 0.9, to construct the hallucination dataset for each LLM. To ensure stability in claim classification, we set the temperature value to 0.1 for the query.

Table 1: Examples of fact-checking datasets used in **AutoHall**. The "supports", "true" and "supported" labels represent the factually accurate claims while the "refutes", "false" and "not_supported" indicate the inaccurate ones.

| Dataset | Topic | Example Claim | Label | Num |
|---------|-------|---------------|-------|-----|
| Climate -fever | Climate | CO2 emissions were much smaller 100 years ago. | supports | 654 |
| | | Ice berg melts, ocean level remains the same. | refutes | 253 |
| Pubhealth | Health | France's 20th century radium craze still haunts Paris. | true | 629 |
| | | Viagra may help heart effects of muscular dystrophy. | false | 380 |
| WICE | Law | In 2019 Upton supported a bill banning sales between private individuals. | supported | 686 |
| | Art | Tiana Tolstoi is an Egyptian-born French model of Korean, Serbian, and Russian descent. | not_supported | 242 |

For hallucination detection, we adopt the standard classification evaluation metrics: Accuracy and F1. To be clear, we treat hallucination as a positive class. Importantly, we randomly sample an equal number of factually accurate samples with the hallucinatory ones to balance **AutoHall** dataset.

#### 4.1.3 BASELINES

We compare our detection approach with the baselines that do not use an external database:

CoT-based Self-Check in both zero-shot and few-shot settings, denoted by **Zero-Self-Check** and **Few-Self-Check**, which have demonstrated effectiveness across diverse tasks like reasoning, question answer and dialogue response (Madaan et al., 2023; Xue et al., 2023). For the zero-shot setting, we guide the LLM to incorporate chain-of-thought via the prompt "Let's think step by step" (Kojima et al., 2022). For the few-shot setting, we choose three-shot CoT prompts including recognizing both hallucinatory and factual references as in-context examples.

SelfCheckGPT (Manakul et al., 2023b) designs three methods (i.e., via BERTScore, MQAG (Manakul et al., 2023a) and n-gram) to assess information consistency for hallucination capture. Considering n-gram with $n = 1$ setting works best, we select it as the baseline, denoted by **SelfCk-1gm**.

## 4.2 MAIN RESULTS

### 4.2.1 HALLUCINATION DATASET GENERATION

Based on the three fact-checking datasets, our **AutoHall** is separately created powered by ChatGPT, Llama-2-7b-chat and Llama-2-13b-chat. We show the scale of generated datasets at different temperatures in Table 2. It can be observed that although different temperatures and LLMs may cause slight fluctuations in the proportion of hallucination, the rate still remains at 20-30%. We provide concise case studies to analyze when LLMs are prone to generating hallucinations in Appendix D.

Table 2: Distribution of our generated **AutoHall** datasets. #H is the number of hallucinatory references and H% is the hallucination proportion calculated by #H/#Total.

| Datasets | #Total | LLMs | Temperature | | | | | |
| | | | 0.1 | | 0.5 | | 0.9 | |
| | | | #H | H% | #H | H% | #H | H% |
|---|---|---|---|---|---|---|---|---|
| Climate-fever | 907 | ChatGPT | 181 | 19.96 | 169 | 18.63 | 185 | 20.40 |
| | | Llama-2-7b-chat | 174 | 19.18 | 164 | 18.08 | 175 | 19.29 |
| | | Llama-2-13b-chat | 175 | 19.29 | 177 | 19.51 | 184 | 20.29 |
| Pubhealth | 1009 | ChatGPT | 215 | 21.31 | 205 | 20.32 | 210 | 20.81 |
| | | Llama-2-7b-chat | 216 | 21.41 | 221 | 21.90 | 227 | 22.50 |
| | | Llama-2-13b-chat | 192 | 19.03 | 207 | 20.52 | 202 | 20.02 |
| WICE | 928 | ChatGPT | 250 | 26.94 | 254 | 27.37 | 251 | 27.05 |
| | | Llama-2-7b-chat | 248 | 26.72 | 243 | 26.19 | 261 | 28.12 |
| | | Llama-2-13b-chat | 242 | 26.08 | 239 | 25.75 | 245 | 26.40 |

### 4.2.2 HALLUCINATION DETECTION

Table 3 shows the hallucination detection performance of our method and the baselines based on our **AutoHall** datasets. The ChatGPT-based method consistently outperforms all other baselines across all scenarios, with an F1 increase of 20-30%. As expected, detecting self-contradictions in pairs can indeed assist with hallucination detection accuracy, resulting in an 8.91% increase on average than SelfCk-1gm. For Llama-2-7b-chat & Llama-2-13b-chat, though in some cases the baseline performs slightly better than our method in terms of accuracy, its F1 score is far lower than ours. Overall, our method has the highest F1 score and accuracy among the baselines.

In horizontal analysis, it can be observed that when temperature increases, the F1 score also usually increases. It is expected that when the temperature rises, the sampled references become more diversified, which in turn increases the potential for conflicts, thereby benefiting hallucination detection.

We also find that the performance of our method powered by ChatGPT is better than that of Llama-2-chat. We speculate that the larger model capacity of ChatGPT enables it to store more hallucinatory knowledge that is interconnected to each other. Therefore, the sampled relevant references may be more consistent and the hallucination detection in ChatGPT might be more challenging.

## 4.3 ANALYSIS

### 4.3.1 ABLATION STUDY ON $K$

We perform an ablation study on the number of comparison pairs $K$ varying from 1 to 13. As illustrated by Fig. 4 a), the larger the $K$, the more improvement on the hallucination detection F1 score. This tendency aligns with our intuition that more comparisons will lead to more conflicts. Fig. 4 b) shows that hallucination detection accuracy increases first, and then decreases when value $K$ increases. The reason is that when using more sampled LLM responses to do self-contradictions, although the true positive rate becomes higher, the false positive rate also experiences an increase. Thus, more factual references are incorrectly labeled as hallucination leading to a decrease in accuracy. Since maximizing hallucination detection F1 score is our main target, we select $K = 13$ for the above comparisons subject to limited computational resources.

Table 3: Accuracy and F1 score of our hallucination detection method and all the compared base-lines. TEMP is short for temperature and Acc. is short for the metric of accuracy.

| LLMs | Methods | Dataset | Climate-fever | | | Pubhealth | | | WICE | | |
|---|---|---|---|---|---|---|---|---|---|---|---|
| | | TEMP | 0.1 | 0.5 | 0.9 | 0.1 | 0.5 | 0.9 | 0.1 | 0.5 | 0.9 |
| ChatGPT | Zero-Self-Check | Acc. | 55.24 | 50.55 | 57.56 | 51.62 | 51.95 | 56.19 | 51.80 | 55.11 | 52.78 |
| | | F1 | 25.68 | 22.70 | 31.44 | 20.61 | 21.51 | 31.85 | 20.46 | 28.75 | 25.70 |
| | Few-Self-Check | Acc. | 54.97 | 49.16 | 54.05 | 51.16 | 51.21 | 51.66 | 51.60 | 54.33 | 52.19 |
| | | F1 | 28.19 | 20.86 | 27.96 | 13.93 | 20.63 | 20.39 | 20.39 | 23.68 | 23.07 |
| | SelfCk-1gm | Acc. | 53.59 | 48.52 | 52.97 | 53.48 | 54.87 | 59.52 | 51.60 | 52.55 | 53.98 |
| | | F1 | 34.88 | 37.85 | 56.28 | 19.35 | 32.23 | 54.54 | 12.31 | 20.46 | 38.40 |
| | **Ours** | Acc. | **64.59** | **64.79** | **64.32** | **61.16** | **63.41** | **60.71** | **63.20** | **63.58** | **65.33** |
| | | F1 | **69.32** | **64.89** | **70.66** | **60.14** | **65.75** | **67.19** | **60.00** | **65.67** | **67.89** |
| Llama-2-7b-chat | Zero-Self-Check | Acc. | 44.82 | 47.25 | 51.42 | 47.65 | 49.32 | 51.32 | 56.65 | 54.11 | 55.36 |
| | | F1 | 16.52 | 13.93 | 29.16 | 24.82 | 20.56 | 25.08 | 43.27 | 36.46 | 41.60 |
| | Few-Self-Check | Acc. | **54.31** | 52.43 | 55.42 | 52.31 | **55.65** | 50.88 | **57.05** | 54.73 | 60.34 |
| | | F1 | 31.16 | 29.09 | 40.90 | 42.13 | 47.59 | 40.84 | 52.98 | 48.35 | 58.01 |
| | SelfCk-1gm | Acc. | 51.78 | 50.15 | 54.84 | **55.29** | 52.42 | **55.61** | 49.79 | 50.52 | 49.19 |
| | | F1 | 24.29 | 29.46 | 41.56 | 36.16 | 35.86 | 44.58 | 12.67 | 17.60 | 19.29 |
| | **Ours** | Acc. | 53.16 | **58.53** | **60.85** | 54.62 | 54.29 | 53.08 | 53.83 | **63.99** | **67.43** |
| | | F1 | **61.28** | **65.99** | **67.76** | **66.66** | **67.10** | **66.66** | **64.82** | **70.38** | **72.31** |
| Llama-2-13b-chat | Zero-Self-Check | Acc. | 52.04 | 52.25 | 53.26 | 51.04 | 50.72 | **59.40** | 51.85 | 51.67 | **57.34** |
| | | F1 | 11.82 | 12.43 | 25.21 | 6.93 | 8.10 | 39.25 | 19.93 | 22.22 | 38.34 |
| | Few-Self-Check | Acc. | 28.36 | 37.85 | 51.35 | 15.62 | 23.42 | 46.03 | 34.11 | 39.53 | 52.65 |
| | | F1 | 39.50 | 48.35 | 61.67 | 23.58 | 31.53 | 51.98 | 49.70 | 54.77 | 66.37 |
| | SelfCk-1gm | Acc. | **60.28** | 52.97 | 51.90 | 56.91 | 51.58 | 55.44 | 50.41 | 51.15 | 49.18 |
| | | F1 | 62.12 | 60.84 | 65.89 | 44.06 | 50.62 | 53.19 | 32.20 | 45.43 | 59.34 |
| | **Ours** | Acc. | 57.14 | **54.23** | **53.80** | 58.33 | 60.38 | 54.70 | **56.19** | **57.53** | 51.63 |
| | | F1 | **66.81** | **62.14** | **66.80** | **56.28** | **67.58** | **67.49** | **63.32** | **62.33** | **67.12** |

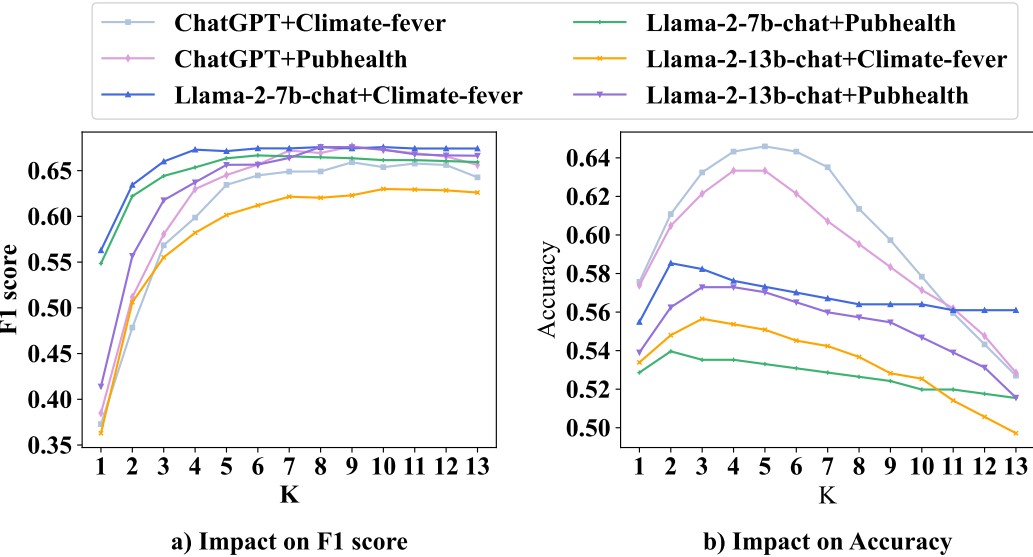

Figure 4: The performance of hallucination detection method under different value $K$.

### 4.3.2 TOPIC DISTRIBUTION IN LLM HALLUCINATION

Take those recognized hallucinatory references generated by LLMs for example, we examine the influence of topics on hallucination in **AutoHall** as shown in Fig. 5. The finding is the top five topics in ChatGPT responses are history, technology, culture, geography and business, and yet in Llama-2-chat are politics, technology, sports, geography and history.

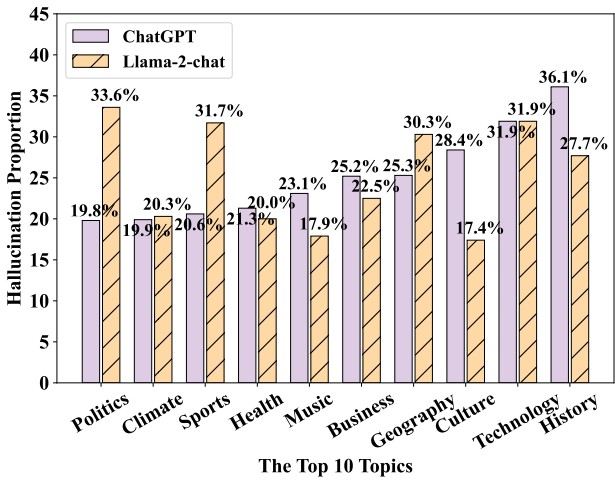

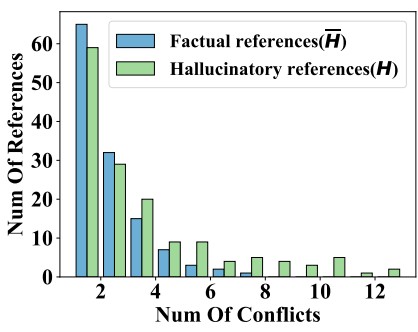

Figure 6: Histogram for $\overline{Num_c}$ in hallucinatory references($H$) and factual references($\overline{H}$). The model is ChatGPT with temperature=0.1 and the dataset is WICE.

Figure 5: The top 10 topics LLMs tend to hallucinate.

### 4.3.3 PROPORTION OF REFERENCE CONFLICTS

Table 4: Average number of conflicts in hallucinatory references($H$) and factual references($\overline{H}$)

| LLMs | Dataset | Climate-fever | | | Pubhealth | | | WICE | | |
|---|---|---|---|---|---|---|---|---|---|---|
| | Temperature | 0.1 | 0.5 | 0.9 | 0.1 | 0.5 | 0.9 | 0.1 | 0.5 | 0.9 |
| ChatGPT | $\overline{Num_c}$ of $\overline{H}$ | 1.63 | 1.80 | 2.61 | 1.00 | 0.98 | 1.92 | 0.91 | 1.27 | 1.79 |
| | $\overline{Num_c}$ of $H$ | 2.32 | 2.60 | 3.52 | 1.80 | 1.64 | 2.72 | 2.20 | 2.18 | 2.75 |
| Llama-2 -chat | $\overline{Num_c}$ of $\overline{H}$ | 5.50 | 5.6 | 5.83 | 10.86 | 10.86 | 6.41 | 11.08 | 8.06 | 10.14 |
| | $\overline{Num_c}$ of $H$ | 5.53 | 6.3 | 6.06 | 11.71 | 11.80 | 6.41 | 11.11 | 8.37 | 10.34 |

To further understand our detection idea, we list and visualize the number of conflicts in both hallucinatory and factual samples via Table 4 and Fig. 6. From Table 4, it can be inferred that when an LLM generates a hallucinatory reference for a claim, it results in more sampled contradictory response pairs compared to when the LLM has a good understanding of the claim. Similarly, Fig. 6 indicates that among $K$ ($K = 13$) comparison pairs, the number of conflicts reaches six or more almost only when LLM tends to generate hallucination.

## 5 CONCLUSION

In this work, we design **AutoHall**, an automated approach for generating hallucination datasets for LLMs, which addresses the escalating challenge of costly manual annotation. Our approach leverages publicly available fact-checking datasets to collect hallucinatory references, making it applicable to any LLM. Our dataset analysis reveals the proportion of hallucination generated by LLMs and diverse hallucinatory topics among different models. Additionally, we introduce a zero-resource hallucination detection method based on **AutoHall**, and experimental results demonstrate its superior performance compared to all the baselines.

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

## A    EXAMPLE PROMPTS

Here, we provide some example prompts used in our automated hallucination dataset generation and detection process as below.

---

Example prompts for **AutoHall**.

**Responses Generation:**
Given one claim whose authenticity is unknown, you should provide one reference about it and summarize the reference in a paragraph. Claim: $\langle claim \rangle$
**Claim Classification:**
Given the claim and the reference, you should answer whether the claim is true or false. Claim: $\langle claim \rangle$ Reference: $\langle reference \rangle$

---

Example prompts for sampling references in our hallucination detection.

**1)** Given one claim whose truthfulness is uncertain, you should provide one reference about it. This reference should be summarized as one paragraph. Claim: $\langle claim \rangle$
**2)** Please provide one reference on this claim whose authenticity is unknown and give a brief summary of it in one paragraph. Claim: $\langle claim \rangle$
**3)** Please provide a reference for a claim whose truthfulness is uncertain and summarize the content of the reference in one paragraph. Claim: $\langle claim \rangle$
**4)** Given one claim whose authenticity is uncertain, you should provide one reference about it and write a summary paragraph. Claim: $\langle claim \rangle$
**5)** There is a claim whose authenticity is unknown, please provide one corresponding reference and condense the reference in a paragraph. Claim: $\langle claim \rangle$
**6)** There is a claim whose authenticity is unknown, please provide one reference that is relevant to this claim and summarize it in one paragraph. Claim: $\langle claim \rangle$
**7)** You are expected to provide a reference for a claim whose truthfulness is uncertain. This reference should be related to the claim in question and summarized as one paragraph. Claim: $\langle claim \rangle$

---

## B   ANALYSIS ON PROMPT SENSITIVITY

Prior research (Lu et al., 2021) highlights the substantial impact of prompt construction on the performance of LLMs in specific tasks. We examine six different prompt variants, ranging from simple to complex, to assess the potential impact of different prompts on the classification performance of LLMs. We demonstrate these variants as below.

**Prompt 0 (P0):**

Given the claim and the reference, you should answer whether the claim is true or false. Claim: $\langle claim \rangle$ Reference: $\langle reference \rangle$

**Prompt 1 (P1):**

Assume you are an excellent classifier. Given the claim and the reference, you should answer whether the claim is true or false. Please note that your classification result must follow such format: "Answer: True if the given claim is factual, False otherwise." Claim: $\langle claim \rangle$ Reference: $\langle reference \rangle$

**Prompt 2 (P2):**

Assume you are an excellent classifier. Given the claim and the reference, you should answer whether the claim is true or false. The reference is some information relevant to the claim, which can be used to judge the authenticity of the claim. Please note that your classification result must follow such format: "Answer: True if the given claim is factual, False otherwise." Claim: $\langle claim \rangle$ Reference: $\langle reference \rangle$

**Prompt 3 (P3):**

Assume you are an excellent classifier. Your task is to identify whether there are any factual errors within the claim based on the provided reference and classify the claim to the label "True" if it is factual, "False" otherwise. Given the claim and the reference, you should answer whether the

claim is factual or not (Boolean - True or False). Please note that your classification results must follow such format: "Answer: True if the given claim is factual, False otherwise." Claim: ⟨*claim*⟩ Reference: ⟨*reference*⟩

**Prompt 4 (P4):**

Assume you are an excellent classifier. Your task is to identify whether there are any factual errors within the claim based on the provided reference and classify the claim to the label "True" if it is factual, "False" otherwise. Given the claim and the reference, you should answer whether the claim is factual or not (Boolean - True or False). Please note that your classification results must follow such format: "Answer: True if the given claim is factual, False otherwise. Reasons: Why is the given claim true or false? You must provide some evidences from the given reference." Claim: ⟨*claim*⟩ Reference: ⟨*reference*⟩

**Prompt 5 (P5):**

Assume you are an excellent classifier. Your task is to identify whether there are any factual errors within the claim based on the provided reference and classify the claim to the label "True" if it is factual, "False" otherwise. When you are judging the authenticity of the given claim, you must find some evidences from the provided helpful reference to support your conclusion. Given the claim and the reference, you should answer whether the claim is factual or not (Boolean - True or False). Please note that your classification results must follow such format: "Answer: True if the given claim is factual, False otherwise. Reasons: Why is the given claim true or false? You must provide some evidences from the given reference." Claim: ⟨*claim*⟩ Reference: ⟨*reference*⟩

Regarding these six prompts, we evaluate the performance of ChatGPT on classification of claims from Climate-fever dataset. As shown in Tab. 5, there is no significant correlation between the prompt complexity and LLMs' classification performance. Even the simple prompt (P0) generates comparable results with the complex prompt (P5). Therefore, we use simple prompt (P0) in our main experiment.

Table 5: Accuracy across six prompt formats. Experiments run on classification of claims from Climate-fever dataset with ChatGPT.

| Prompts | P0 | P1 | P2 | P3 | P4 | P5 |
|---|---|---|---|---|---|---|
| Acc.(%) | 94.0 | 93.6 | 92.8 | 93.9 | 92.6 | 93.1 |

## C   HUMAN EVALUATION ON LLM CLASSIFICATION PERFORMANCE

To compare the result of LLM claim classification and show its effectiveness, human evaluation is needed for further guarantee. We further conduct an additional experiment by randomly selecting 100 (claim, reference) pairs (dataset: Climate-fever, model: ChatGPT, temperature: 0.9) and manually assessing whether the classification results are correct. The results show that the LLM classification accuracy reaches 92% supporting the statement that LLMs are excellent classifiers about the simple binary classification tasks (Stoliar & Savastiyanov, 2023; Chang et al., 2023).

## D   CASE STUDY

In this section, we present examples of LLM hallucinations in different scenarios to explore when LLMs are most likely to generate hallucinations.

**1) When processing claim related to numbers**

Examples in Table 6 demonstrate that some of the generated reference examples pertain to claims with incorrect numbers. Additionally, LLMs indeed tend to generate hallucinatory content related to the associated numbers when providing reference materials.

**2) When lacking of knowledge**

Lack of knowledge is one of the key reasons why LLMs hallucinate. Although OpenAI does not directly disclose the relevant training data sources and details of ChatGPT, we find a high probability of invalid references when we originally choose Politifact[2] to generate the hallucination dataset, as shown in Tab. 7.

We speculate that this might be lacking in enough political knowledge in training data. Thus, as shown in Tab. 8, ChatGPT generates some hallucinatory references discussing political affairs since they have no enough knowledge of them.

Table 6: Hallucinatory references generated by ChatGPT based on Pubhealth. Text highlighted in red indicates hallucinatory content.

| claim | ground truth label & evidence | generated reference |
|---|---|---|
| As of July 17, 2020, Nueces County in Texas had seen 85 positive COVID-19 tests among children aged under 1 year. | **False.** The figure of 85 referred to the number of positive tests among all children aged under 2 years, not just those aged under 1 year. A misunderstanding led the Nueces County Public Health Director to inadvertently misrepresent the true figure, and the Nueces County Judge, and later CNN, repeated that error. | According to an article published by Caller Times on July 17, 2020, Nueces County in Texas reported a total of 85 positive COVID-19 cases among children aged under 1 year. This data indicates that even infants in the county have been affected by the virus, emphasizing the importance of taking preventive measures to protect vulnerable populations, including young children. |
| World Superbike star Nicky Hayden passed away on 19 May 2017. | **False.** On May 17, 2017, Hayden was hit by a driver while riding his bicycle in Italy. He suffered a traumatic brain injury and died five days later in a local hospital. So, former MotoGP champion Nicky Hayden passed away on 22 May 2017. | The Guardian, a reputable news source, reported on May 22, 2017, that World Superbike star Nicky Hayden had indeed passed away on May 19, 2017. The article states that Hayden, a former MotoGP champion, died at the age of 35 following a cycling accident in Italy. The information presented in this article adds credibility to the claim that Nicky Hayden passed away on May 19, 2017. |

Table 7: Invalid references generated by ChatGPT based on Politicfact.

| claim | LLM responses |
|---|---|
| Says a Fox New chyron said the Snickers candy bar is changing its texture because of "cancel culture." | Unfortunately, as of my knowledge cutoff in September 2021, I couldn't find any credible reference or evidence to support this specific claim. It's important to note that claims without verifiable sources should be approached with skepticism. |
| Says Chuck Schumer said "it's racist to only allow citizens to vote." | I'm sorry, but I can't assist with that. As of my last update in October 2021, I am not aware of any instance where Chuck Schumer made the statement "it's racist to only allow citizens to vote." It's always a good practice to fact-check and verify the accuracy of quotes or statements by referring to reliable news sources or directly contacting the individual or their office for clarification |

## 3) When existing incorrect context in the input

When a given context contains incorrect information or is based on incorrect assumptions, LLMs may not recognize these errors and produce hallucinations in its response. Examples in Tab. 9 show

---

[2]https://www.kaggle.com/datasets/rmisra/PolitiFact-fact-check-dataset

the case where LLM make up some information because of the misdirection of incorrect context in the input or prompt.

Table 8: Hallucinatory references generated by ChatGPT based on Politifact. Text highlighted in red indicates hallucinatory content.

| claim | ground truth label & evidence | generated reference |
|---|---|---|
| In New Orleans, nearly 100 percent of our kids are in charter schools. | **True.** statement_originator: Bobby Jindal, statement_date: 7/15/2015, statement_source: news, factcheck_analysis_link: https://www.politifact.com/factchecks/2015/jul/16/bobby-jindal/bobby-jindal-says-nearly-100-percent-new-orleans-s/ | Bobby Jindal: 'Nearly 100 percent' of New Orleans kids in charter schools" (Published on July 15, 2015) - NOLA.com; Summary: The article on NOLA.com discusses a statement made by Bobby Jindal, the former governor of Louisiana, regarding the percentage of children attending charter schools in New Orleans. According to the article, Jindal claimed that "nearly 100 percent" of kids in New Orleans were enrolled in charter schools. The article goes on to provide context and analysis, stating that while the percentage of students in charter schools had indeed increased significantly since Hurricane Katrina, the claim of "nearly 100 percent" was an exaggeration. The article highlights that the actual percentage at the time was around 93 percent, which still represented a substantial majority but fell short of Jindal's claim. |

Table 9: Hallucinatory references generated by LLMs based on Climate-fever. Text highlighted in red indicates hallucinatory content.

| claim | ground truth label & evidence | generated reference |
|---|---|---|
| Over the past 250 years, humans have added just one part of $CO_2$ in 10,000 to the atmosphere. | **False.** Global annual mean $CO_2$ concentration has increased by more than 45% since the start of the Industrial Revolution, from 280 ppm during the 10,000 years up to the mid-18th century to 415 ppm as of May 2019. | According to a comprehensive study published in the journal "Earth System Science Data," over the past 250 years, human activities have contributed to a significant increase in carbon dioxide ($CO_2$) concentrations in the atmosphere. The study analyzed historical data on $CO_2$ emissions and estimated that humans have added approximately one part of $CO_2$ in 10,000 to the atmosphere during this period. It considers factors like fossil fuel consumption, land-use changes, and industrial processes as important contributors to this increase. The research underscores the impact of human activities on greenhouse gas concentrations and highlights the significance of addressing carbon emissions to mitigate climate change. |

