# OpenReview forum: "AutoHall: Automated Hallucination Dataset Generation for Large Language Models"
_ICLR.cc/2024/Conference — Submitted to ICLR 2024_

### Official Review · Reviewer_n69Q · 2023-10-30

**Soundness:** 2 fair
**Presentation:** 3 good
**Contribution:** 1 poor
**Rating:** 3
**Confidence:** 4

**Summary:**

In response to the challenge of labor-intensive and costly manual annotation processes for determining the hallucinatory nature of texts generated by Language Model Models (LLMs), this paper introduces an approach that automates the creation of model-dependent hallucination datasets by leveraging existing fact-checking datasets. For each claim (from the fact-checking dataset), an LLM is asked to generate the corresponding references to the claim and predict whether the claim is correct or not. If the LLM’s prediction is not consistent with the ground truth, the generated references are labeled as hallucination. To identify hallucinations, the authors present a two-step procedure: first, they prompt the LLM to generate multiple references in relation to a given claim, and subsequently, the same LLM is used to assess whether any inconsistencies or contradictions exist among the generated references. The presence of such self-contradictions is indicative of the likely hallucinatory nature of the claim.

**Strengths:**

(1) The paper presents an approach for the automatic generation of model-specific hallucination datasets through the utilization of existing fact-checking datasets.

(2) The paper is well-written with clarity and comprehensibility.

**Weaknesses:**

(1) The authors place considerable emphasis on the model-specific nature of the automatically constructed hallucination datasets. It is unclear why the creation of such datasets is necessary, as they may not be applicable for evaluating other models due to their model-specific attributes. Furthermore, the paper does not propose a method for leveraging these datasets to mitigate the generation of hallucinatory content by the models.

(2) The paper lacks a clear and coherent explanation of the interrelation between the two methods presented, namely hallucination dataset construction and hallucination detection.

(3) The paper's reliance on the concept of self-contradictions as a novel approach for detecting hallucination in LLMs is questionable. Several prior studies have explored similar methodologies in hallucination detection, raising concerns about the novelty of the proposed method.

**Questions:**

(1) Could you explain the relationship or connection between the two methods introduced in this study, namely, hallucination dataset construction and hallucination detection?

(2) What is the rationale behind relying on self-generated datasets for evaluating the efficacy of the proposed detection methods? Would it not be more compelling to employ other datasets for evaluation purposes?

(3) What is the underlying objective behind creating model-specific hallucination datasets, and how were these datasets applied in this study?

---

> ### Author Response · Authors · 2023-11-18
>
> Thanks for your feedback.
>
> **C#1&Q#3: What is the underlying objective behind creating model-specific hallucination datasets, and how were these datasets applied in this study?**
>
> We clarify the necessity of model-specific hallucination datasets in the **Sec.2** of our paper. Due to the scarcity of high-quality hallucination datasets, it is challenging to determine the specific types and degrees of hallucination exhibited by different LLMs. Existing approaches for hallucination detection often require manual annotation, which is time-consuming and labor-intensive.
> Our objective is to provide a unified method for automatically collecting such datasets applicable to various models. These datasets serve as a benchmark for evaluating different LLMs and facilitate the development of techniques to detect and mitigate LLM hallucinations. Researchers can leverage our AutoHall to collect up-to-date model-specific hallucination datasets, even as models update.
>
> **C#1: The paper does not propose a method for leveraging these datasets to mitigate the generation of hallucinatory content by the models.**
>
> Hallucination mitigation is beyond the scope of this paper and our focus is automatic hallucination detection dataset construction and hallucination detection based on our constructed datasets.
>
> **C#2&Q#1: Could you explain the relationship or connection between the two methods introduced in this study, namely, hallucination dataset construction and hallucination detection?**
>
> Sure. We first need construct up-to-date hallucination datasets to find the varying characteristics of hallucination among different LLMs, which is the first part of our study. And then in the second part, we design a novel hallucination detection approach based on our constructed datasets.
>
> **C#3: Concerns about the novelty of the proposed method.**
>
> Our work makes a significant contribution in an automatic approach for generating hallucination detection datasets, which aligns with the title of our paper and has enough novelty. In addition, proposed hallucination detection method is another contribution. Although several prior studies utilize self-contradictions for detection, we employ the LLM itself to assess the presence of contradictions instead of relying on external tools (Manakul P et al., 2023). And unlike previous methods that compare the presence of contradictions among multiple sampled responses, we adopt a pairwise comparison approach to reduce false negatives.
>
> Refs:
>
> 1. Manakul P, Liusie A, Gales M J F. Selfcheckgpt: Zero-resource black-box hallucination detection for generative large language models[J]. arXiv preprint arXiv:2303.08896, 2023.
>
> **Q#2: What is the rationale behind relying on self-generated datasets for evaluating the efficacy of the proposed detection methods? Would it not be more compelling to employ other datasets for evaluation purposes?**
>
> Here are some reasons why we do not employ other datasets for evaluation:
>
> 1.	Applicability of existing hallucination datasets: Some of the publicly available hallucination datasets were collected from the previous version of LLMs such as GPT-3. However, these datasets are not suitable for evaluating the latest model which may produce different types or degrees of hallucination.
>
> 2.	Differences between task-specific datasets and our open-domain generation scenario: Some existing datasets are task-specific (such as for summarization or QA tasks), while our work focuses on hallucination detection in open-domain generation scenarios. Those datasets may not adequately cover the range of hallucinations that occur in open-domain generation contexts.

---

### Official Review · Reviewer_AdN8 · 2023-10-31

**Soundness:** 3 good
**Presentation:** 2 fair
**Contribution:** 2 fair
**Rating:** 3
**Confidence:** 4

**Summary:**

The paper introduces AutoHall, a method to automatically generate datasets to test models for hallucination. To do so, AutoHall takes claims from an existing fact checking dataset, then asks the LLM to generate “references” that support the claims (although the references seem like a justification in text). Then, for each claim and reference, AutoHall asks a LLM whether the claim is true or false according to the reference. And finally, if the claim is false according to the reference but the ground truth label is true (or vice versa), we know there’s hallucination (either in the claim classification stage or the reference generation stage). Finally, to detect hallucination, the authors test whether the reference contradicts an original reference (form the original dataset). They find that ChatGPT, Llama-2-7b-chat and Llama-2-13b chat all hallucinate at roughly the same rate, and that their detection method outperforms existing detection methods.

**Strengths:**

* This paper offers a nice way to repurpose existing datasets to test for hallucination
* The detection results seem encouraging, especially over other methods.

**Weaknesses:**

* The hallucination this paper tests for could be out-of-distribution for the hallucination we care about in practice. For example, hallucinations measured by AutoHall could be failures of the LLM classifier (for detecting whether something is supported). Even generating supportive content for statements seems like a task these LLMs are rarely used for in practice.
* One selling point of AutoHall is that it's automatic, but it requires relying claims and ground-truth evidence from fact checking datasets, which were curated by humans.
* The reported hallucination numbers, where Llama and ChatGPT produce similar hallucination rates, are very different than the numbers reported in https://arxiv.org/abs/2305.14251where ChatGPT is nearly twice as factual as fine-tuned versions of Llama (though it's LLama 1 instead of 2, I'd be surprised if Llama 2 closes the gap), raising questions about the fidelity of the generated datasets.

**Questions:**

* Why do you think there's the discrepency in the AutoHall conclusions and the factscore conclusions?

---

> ### Author Response · Authors · 2023-11-18
>
> Thanks for your review.
>
> **C#1: The hallucination this paper tests for could be out-of-distribution for the hallucination we care about in practice. For example, hallucinations measured by AutoHall could be failures of the LLM classifier (for detecting whether something is supported). Even generating supportive content for statements seems like a task these LLMs are rarely used for in practice.**
>
> We need clarify our focus on generating references about claims, i.e., finding some relevant evidences and we have emphasized `claim whose authenticity is unknown` in the prompt used. The statement `prompt a model to generate supportive references` in Sec. 3.2 used the inappropriate word `supportive` and we will revise it to the word `relevant` in the later version.
>
> Moreover, in various argumentation-based tasks, such as information retrieval and argumentation mining, the availability and reliability of diverse references is crucial for comprehensive analysis and evaluation. Therefore, our work can help the hallucination detection of evidences provided in such contexts.
>
> We further conduct an additional experiment by randomly selecting 100 (claim, reference) pairs (**dataset: Climate-fever, model: ChatGPT, temperature: 0.9**) and manually assessing whether the classification results are correct. The results show that the LLM classification accuracy reaches over **92%** supporting the statement that LLMs are excellent classifiers about the simple binary classification tasks (Stoliar al., 2023; Yang et al., 2023; Chang et al., 2023). So, we assume that hallucinations occur in reference texts that lead to misclassification of claims.
>
> Refs:
>
> 1. Stoliar et al., Using LLM classification in foresight studies. 2023.
>
> 2. Yang et al., Large language models can rate news outlet credibility. 2023.
>
> 3. Chang et al., A Survey on Evaluation of Large Language Models. 2023.
>
> **C#2: One selling point of AutoHall is that it's automatic, but it requires relying claims and ground-truth evidence from fact checking datasets, which were curated by humans.**
>
> We politely disagree with your statement. Publicly available fact-checking datasets are already with labels, and our AutoHall process does not involve any manual annotation. Considering current hallucination benchmarks are highly dependent on human annotation (Li et al., 2023; Umapathi et al., 2023; Dale et al., 2023), our automatic AutoHall approach is undoubtedly meaningful.
>
> Refs:
>
> 1. Li et al., Halueval: A large-scale hallucination evaluation benchmark for large language models. EMNLP 2023.
>
> 2. Umapathi et al., Med-halt: Medical domain hallucination test for large language models. EMNLP 2023.
>
> 3. Dale et al., HalOmi: A Manually Annotated Benchmark for Multilingual Hallucination and Omission Detection in Machine Translation. 2023.
>
> **C#3: The reported hallucination numbers, where Llama and ChatGPT produce similar hallucination rates, are very different than the numbers reported in https://arxiv.org/abs/2305.14251 where ChatGPT is nearly twice as factual as fine-tuned versions of Llama (though it's Llama 1 instead of 2, I'd be surprised if Llama 2 closes the gap), raising questions about the fidelity of the generated datasets.**
>
> We have searched for some relevant works to demonstrate our conclusion. (Du et al., 2023) state that GPT-3.5-turbo has a hallucination rate of 26% in Fig. 3 which is consistent with ours. Meta claims (A.4.8, P69, Touvron et al., 2023) Llama-2-chat indeed improves a lot in terms of LLM hallucinations, which is also demonstrated in (Fig.1 & Tab. 1, Cheng et al., 2023).
>
> Refs:
>
> 1. Du et al., Quantifying and Attributing the Hallucination of Large Language Models via Association Analysis. 2023.
>
> 2. Touvron et al., Llama 2: Open foundation and fine-tuned chat models. 2023.
>
> 3. Cheng et al., Evaluating Hallucinations in Chinese Large Language Models. 2023.
>
> **Q#1: Why do you think there's the discrepancy in the AutoHall conclusions and the factscore conclusions?**
>
> Factscore evaluates traditional fact-checking tasks doing information retrieval on existing claims. However, hallucination detection emphasizes evaluation on the content generated by the models themselves, on which traditional fact-checking tasks pay little attention.
> Besides, the FactScore evaluation is not equivalent to the hallucination rates concluded by our AutoHall. Fine-grained FactScore focuses on atomic facts which are more fundamental units than sentences, while the hallucination rate is a sentence-level metric. Therefore, the conclusions drawn from AutoHall and FactScore evaluations are not directly comparable due to the above differences.

---

### Official Review · Reviewer_ud13 · 2023-11-03

**Soundness:** 2 fair
**Presentation:** 2 fair
**Contribution:** 2 fair
**Rating:** 5
**Confidence:** 4

**Summary:**

This paper studied the critical hallucination problem of large language models and proposed AutoHall, an automatic method for constructing model-tailored hallucination datasets. AutoHall utilizes fact-checking datasets, prompts LLMs to produce reference outputs from given claims, and subsequently determines the correctness of these references using the verified ground-truth labels. Leveraging the datasets constructed via AutoHall, the authors developed a hallucination detection technique, which relies on identifying inconsistencies across multiple generated references to predict hallucinations. The efficacy of this framework was tested on popular models such as ChatGPT and LLaMA-2-chat, showcasing its superiority over several established zero-resource detection methods.

**Strengths:**

* This paper took a step towards addressing a significant and interesting problem, the hallucination evaluation of LLMs with no assumption of external resources, which is more suitable for real scenarios.
* The automatic data creation method is well-motivated by timeliness and transferability problems of current model-specific and manually created datasets.
* The authors demonstrated the effectiveness of their framework by comparing several previous baseline models and also gave some analysis of the properties of hallucinatory references, which would benefit future research in this direction.

**Weaknesses:**

Though this paper is well-motivated, there are three main problems:

* Some designs of the data generation and detection methods are questionable. In detail:
  1. The method to classify the generated references is unreasonable since the two-phase pipeline brings much noise and accumulates errors. In Step 2, LLMs take the risk of producing wrong answers from factual references and vice versa.
  2. The problem caused by model randomness in the data generation process has not been considered. For example, for a given (claim, reference) pair, the LLM’s label prediction could be influenced by sampling-based decoding and the form of instructions, as LLMs are sensitive to different instructions (Zhou et al., 2023).
  3. The necessity of model-specific generation has not been verified. This motivation makes sense, but the authors should also justify it by testing the transferability of datasets generated by different models.

* The proposed pipeline lacks novelty.
   1. The method of automatically generating hallucinated references was proposed by (Agrawal et al., 2023). The only difference lies in that the authors generated references from existing fact-checking claims instead of topic words.
   2. The proposed self-contradiction-based hallucination detection method has also been proposed in previous work (Mündler et al., 2023; Manakul et al., 2023; Cohen et al., 2023), among which Mündler et al., and Cohen et al., both utilized LLMs as the contradiction judge. Besides, the design of self-contradiction detection in $(Y,Y^{'}_k)$ pair seems meaningless. Since the generation process of $Y$ is the same as that of $Y^{'}_k$, $Y$ can be considered one of $Y^{'}_k$. The claim `*avoid the problem that SelfCheckGPT incorrectly identifies the conflicts in the K sentences generated*' was not verified.

* The experimental settings are not sound enough, making results unconvincing.
   1. As mentioned above, the quality of generated datasets is not guaranteed, and hence the conclusion is also questionable.
   2. Some essential baselines are not compared. For example, (Mündler et al., 2023), which also uses LLMs as the contradiction judge in an NLI manner.
   3. Since the contradiction judge based on LLMs also introduces randomness and noise, human evaluation is needed for further guarantee.

* Some essential designs need more in-depth analysis. a) Why did self-check perform worse in the few-shot setting than the zero-shot setting with ChatGPT in Table 3? b) Why was 1gm worse than the few-shot with LLaMa2-7B?

Refs:
* Zhou et al., Large Language Models Are Human-Level Prompt Engineers. ICLR 2023.
* Agrawal et al., Do Language Models Know When They’re Hallucinating References? 2023.
* Cohen et al., LM vs LM: Detecting Factual Errors via Cross Examination. 2023.
* Manakul et al., SELFCHECKGPT: Zero-Resource Black-Box Hallucination Detection for Generative Large Language Models. 2023.
* Mündler et al., Self-contradictory Hallucinations of Large Language Models: Evaluation, Detection and Mitigation. 2023.

**Questions:**

1. Fig. 1 is blurry.

2. Is Fig. 6 the opposite of the conclusion? The more conflicts there are, the more factual references.

---

> ### Author Response · Authors · 2023-11-18
>
> Thanks for your meticulous comments.
>
> **C#1: The method to classify the generated references is unreasonable.**
>
> Numerous studies (Li et al., https://arxiv.org/abs/2305.14623; Cheung et al., https://arxiv.org/abs/2309.00240; Zhang et al., https://arxiv.org/abs/2304.03728) have demonstrated the high accuracy of LLM-based fact-checking approaches despite the potential for classification errors, among which Li et al. and Zhang et al. also used LLMs with instructions like ours.
> We further conduct an additional experiment by randomly selecting 100 (claim, reference) pairs (**dataset: Climate-fever, model: ChatGPT, temperature: 0.9**) and manually assessing whether the classification results are correct. The results show that the LLM classification accuracy reaches **92%**.
>
> **C#2: The problem caused by model randomness in the data generation process has not been considered.**
>
> In our study, we carefully design and experiment with the prompts used for classification, and the prompt ultimately selected prove to be effective across different models.
> We also conduct additional experiments using **six** prompt variants (from simple (P0) to complex (P5)) to assess the classification performance on prompt sensitivity. Results show that there is no significant correlation between the prompt complexity and LLMs’ classification accuracy.
> | Prompts | P0 | P1 | P2 | P3 | P4 | P5 |
> | ---- | ---- | ---- | ---- | ---- | ---- | ---- |
> | Acc.(%) | 94.0 | 93.6 | 92.8 | 93.9 | 92.6 | 93.1|
>
>
> **C#3: This motivation makes sense, but the authors should also justify it by testing the transferability of datasets generated by different models.**
>
> The transferability of datasets generated by different models is beyond the scope of our study and it is improper to test whether other models have hallucinations using data generated by a specific model. It is important to note that our research aims to automatically construct hallucination datasets and explore what types or topics of LLM responses that tend to be hallucinatory.
>
> **C#4: The method of automatically generating hallucinated references was proposed by (Agrawal et al., 2023.)**
>
> Not really. Our AutoHall approach centers on claim classification based on the idea that generated hallucinatory references lead to misclassification of claims since LLMs are excellent classifiers (Stoliar et al., Using LLM classification in foresight studies). This is totally different from direct queries in (Agrawal et al., 2023) by prompting the LM to check the existence of the reference.
>
> **C#5: The design of self-contradiction detection in pair seems meaningless. & Q#2: About Fig.6.**
>
> Not really. Self-contradiction detection in $(Y,Y_{k}^{'})$ pair is meaningful as it avoids the situation that detecting conflicts together like SelfCheckGPT incorrectly labels hallucination in Y when in fact hallucinations exist among K sentences. We admit that there is still a possibility of hallucination existing in $Y^{'}_{k}$ while not in Y which increases false positive to some extent. However, by doing self-contradiction detection in pairs, we can find almost only when hallucinations exist in Y can the number of conflicts exceed half from Fig. 6. Thus, our approach in general achieves higher F1 score than SelfCheckGPT as is shown in our experiments. And we wrote the legend text in reverse order by mistake in Fig. 6. We will revise it in the later version.
>
> **C#6: Some essential baselines are not compared. For example, (Mündler et al., 2023).**
>
> Not really. Although the work (Mündler et al., 2023) also uses self-contradictions for detection, they are dependent on context (blue text named description in Fig. 2) which is different from our open-domain generation scenarios without any context guided.
>
> **C#7: a) Why did self-check perform worse in the few-shot setting than the zero-shot setting with ChatGPT in Tab. 3? b) Why was 1gm worse than the few-shot with LLaMa2-7B?**
>
> a)	The experimental results in (Tab. 5, Chern et al., https://arxiv.org/abs/2307.13528) also use the same Self-Check baselines and show the same comparison results with ours. We suppose that CoT mechanism may not perform its best in open-domain generation scenarios, thus few-shot or zero-shot setting have little impact. So regardless of the types of LLMs, Self-Check baselines are prone to false positive and perform worse.
>
> b)	The performance of SelfCk-1gm relies highly on the variety of LLM responses. Since the number of parameters in a model determines its language understanding and generation capabilities (Tab.3, Touvron et al., https://arxiv.org/abs/2307.09288), the 7B model may be the weakest in performance among baselines. Less world knowledge, less variety in responses and more difficult to detect inconsistency, thus SelfCk-1gm with LLaMa2-7B may be worse than the few-shot while with other baselines are the opposite.
>
> **Q#1: Fig. 1 is blurry.**
>
> Thanks for pointing this detail out. We will revise it in the later version.

---

> ### Comment · Reviewer_ud13 · 2023-11-23
> **Thanks for the additonal experiments**
>
> Thank you for answering my questions and providing additonal experiments. I think my first two concerns, C1 (validity of the classification method for generated references) and C2 (randomness in the data generation process), have been addressed. However, the other three main concerns remain.
>
> 1.  Necessity of model-specific generation. This is a strong claim and one of the core motivations in this work. The reason we need automatically generated datasets is "each model requires a full annotation of the dataset...such a dataset
> is also time-sensitive"  as mentioned in Sec.1. If the dataset has a good transferability, which means one dataset is effective to evaluate diverse LLMs, then we don't need such a generation algorithm. One crowd-sourced dataset is effective forever. The authors should justify that some static datasets are not suitable for other LLMs. Or, datasets generated by one LLM doesn't work for another.
>
> 2. The novelty concern, especially on self-contradiction-based hallucination detection method. The essence of the proposed method still lies in utilizing the consistency among the generated references. The only difference from previous work is whether to let the LLM judge all Ys at once or to do so one by one. Moreover, the improvement in detection effectiveness brought about by this modification has not been verified through ablation study.
>
> 3. The performance of the proposed self-contradiction-based hallucination detection method hasn't been verified by human annotators.
>
> Therefore, I  raised my score to 5 but not higher.

---

### Meta-Review · Area_Chair_ydj4 · 2023-12-06

**Metareview:**

This paper presents an approach called AutoHall for automatically generating hallucination datasets for different LLMs by using existing fact-checking datasets, and then introduces a zero-resource hallucination detection method based on AutoHall. Experimental results demonstrate the prposed detection method outperforms the baselines.

Strengths: The problem investigagted in this paper is interesing and important. The proposed dataset construction and detection methods sound reasonable. The paper is easy to read.

Weaknesses: The novelty of the proposed hallucination detection method based on self-contradiction is quite limited, as several prior studies have explored similar methodologies in hallucination detection.  It is unclear why the creation of model-specific datasets is necessary, as they may not be applicable for evaluating other models due to their model-specific attributes. It would be more reasonable to evaluate different LLMs on the same datasets, and thus we can fairly compare the capability of different LLMs. Lastly, the performance of the proposed hallucination detection method needs to be verified by human annotators.

Considering the weaknesses outweigh the strengths, I recommend to reject this paper.

**Justification For Why Not Higher Score:**

see the meta-review.

**Justification For Why Not Lower Score:**

see the meta-review.

---

### Decision · Program_Chairs · 2024-01-16

Reject